# Locally Free Weight Sharing for Network Width Search

**Xiu Su[1], Shan You[2,3]\*, Tao Huang[2], Fei Wang[2], Chen Qian[2], Changshui Zhang[3], Chang Xu[1]**
[1]School of Computer Science, Faculty of Engineering, The University of Sydney, Australia
[2]SenseTime Research
[3]Institute for Artificial Intelligence, Tsinghua University (THUAI)
Beijing National Research Center for Information Science and Technology (BNRist)
Department of Automation, Tsinghua University, Beijing, P.R.China
`xisu5992@uni.sydney.edu.au, youshan@sensetime.com,`
`{huangtao,wangfei,qianchen}@senseauto.com,`
`zcs@mail.tsinghua.edu.cn, c.xu@sydney.edu.au`

## Abstract

Searching for network width is an effective way to slim deep neural networks with hardware budgets. With this aim, a one-shot supernet is usually leveraged as a performance evaluator to rank the performance w.r.t. different width. Nevertheless, current methods mainly follow a manually fixed weight sharing pattern, which is limited to distinguish the performance gap of different width. In this paper, to better evaluate each width, we propose a loCAlly FrEe weight sharing strategy (CafeNet) accordingly. In CafeNet, weights are more freely shared, and each width is jointly indicated by its base channels and free channels, where free channels are supposed to locate freely in a local zone to better represent each width. Besides, we propose to further reduce the search space by leveraging our introduced FLOPs-sensitive bins. As a result, our CafeNet can be trained stochastically and get optimized within a min-min strategy. Extensive experiments on ImageNet, CIFAR-10, CelebA and MS COCO dataset have verified our superiority comparing to other state-of-the-art baselines. For example, our method can further boost the benchmark NAS network EfficientNet-B0 by 0.41% via searching its width more delicately.

## 1 Introduction

Deep neural networks are easily constructed by stacking multiple layers of non-linearities on top of each other. But it is a much more delicate job than it seems. Before going into the deep, we have to determine the width of each layer[1] (Chen et al., 2020) to be stacked, especially given the consideration of the computational budgets (He et al., 2016; Han et al., 2015; Ioffe & Szegedy, 2015; Huang et al., 2017) (*e.g.*, FLOPs, and latency) and the resulting network performance. The width of a pretrained over-parameterized neural network can be further trimmed by classical channel pruning techniques (Liu et al., 2019b; He et al., 2017; Tang et al., 2019) to eliminate the redundancy. Inspired by the spark of neural architecture search (NAS) (Liu et al.; Cai et al., 2019; Hu et al., 2020; Yang et al., 2020; Huang et al., 2020; Yang et al., 2021), recent methods attempt to directly search for the optimal network width for a given network, and have achieved remarkable performance in various FLOPs levels or acceleration rates, such as AutoSlim (Yu & Huang, 2019), MetaPruning (Liu et al., 2019a) and TAS (Dong & Yang, 2019). Then the obtained models can be deployed in edge devices, or even boosted by other compact techniques, such as quantization and knowledge distillation Hinton et al. (2015); You et al. (2017); Kong et al. (2020); Du et al. (2020).

To determine the optimal network width, we need to evaluate and compare the performance of different network width. One of the most straightforward yet authentic approaches is to examine their training-from-scratch performance and traverse all possible settings of network width (not exceeding a maximum allowable width). However, exhaustive training is computationally unaffordable due to

---

\*Corresponding author.
[1]Other literature also use the number of channels/filters to indicate the network width.

| (a) Toy example | (b) Channel assignment of sub-network w.r.t. differernt weight sharing patterns |

Figure 1: (a) A toy example of the supernet. We only search for the width for its one convolutional layer with all 6 channels. (b) Examples of different weight sharing patterns. For the fixed weight sharing pattern, leftmost $c$ channels are assigned for the sub-network of width $c$. For the locally free weight sharing pattern, the offset of free zone is set to $r = 1$.

the huge search space. Taking MobileNetV2 as an example, 100 channels in each of the 25 layers result in $100^{25}$ possible network architectures. For sake of searching efficiency, current methods Chen et al. (2020); Yu & Huang (2019); Dong & Yang (2019) usually follow a weight sharing strategy by leveraging a one-shot supernet, with various sub-networks sharing the same weights with the supernet. Then each network width can be efficiently evaluated by querying the performance of its corresponding sub-network in the supernet.

In this way, how to specify the sub-network for each network width indicates the search space and matters for the performance evaluation. Without loss of generality, we count channels in a layer from the left. A popular weight sharing strategy for the supernet then follows a fixed weight sharing pattern, which simply assigns the left $c$ channels as the sub-network for the width $c$ (see the gray dots in Fig.1(b)). In this manner, the search space of a layer with $n$ channels is $\mathcal{O}(n)$. However, this fixed pattern imposes an inherent constraint on the search space. Any two sets of width fully share their weights, limiting the accurate evaluation ability of supernet and affecting the searching performance accordingly. Ideally, each channel from the supernet shall enjoy the freedom to be selected, *i.e.*, any channel has a probability to occur in the subnetwork, no matter it locates at the leftmost or the rightmost of a layer. But the price of this *full* channel freedom is the dramatic increase of the search space size to $\mathcal{O}(2^n)$, which affects the feasibility of width search. Instead of this unaffordable full freedom, we prefer restrictions and regularization over the search space to keep the channel freedom within an acceptable range, which benefits the efficiency of width search.

In this paper, we propose a loCAlly FrEe weight sharing strategy of the supernet (*i.e.*, CafeNet) for network width search. For each width $c$, our goal is to specify a sub-network within supernet by assigning $c$ channels, indicated by an index set $\mathcal{I}(c)$. To balance the weight sharing freedom and the searching efficiency, we split the index set $\mathcal{I}(c)$ into two parts, *i.e.*, $c_b$ *base channels* $\mathcal{I}_b(c)$ and $c_f$ *free channels* $\mathcal{I}_f(c)$, with $\mathcal{I}(c) = \mathcal{I}_b(c) \cup \mathcal{I}_f(c)$ and $c = c_b + c_f$. (see Fig.1(b)). In detail, base channels follow the fixed weight sharing pattern, while free channels are encouraged to select freely from a given zone, which is the neighborhood of the $c$-th channel. In this way, each width $c$ can be better evaluated by multiple locally free sub-subnetworks. The supernet CafeNet can be trained by optimizing the sub-network with the minimum loss for each width and search for the one with the maximum accuracy. To further ease the burden of searching cost, we take a group of channels (*i.e.*, *bin*) as the minimal searching unit. The bin sizes are carefully set by encouraging FLOPs to be more evenly allocated over bins. We conduct extensive experiments on the benchmark ImageNet (Deng et al., 2009) and CIFAR-10 (Krizhevsky et al., 2014) datasets to validate the superiority of our proposed CafeNet. The effectiveness of our CafeNet is also examined on the face attribute recognition task on CelebA (Liu et al., 2015) dataset and the transferability of our searched backbone for object detection task on MS COCO dataset (Everingham et al., 2010).

## 2 PROBLEM FORMULATION

With a given network structure, the aim of the network width search is to optimize its width for each layer under certain budgets. Suppose the target network has $L$ layers, then the network width can be represented as a $L$-length tuple of layer-wise width $\boldsymbol{c} = (c_1, c_2, \ldots, c_L)$ in which $c_i$ represents the width at the $i$-th layer. Without loss of clarity, we simply focus on searching for the optimal layer width of $c_i$. Denote the maximum allowable width at the $i$-th layer as $n_i$. Then the number of all sub-networks in the supernet amounts to $\prod_{i=1}^{L}(2^{n_i} - 1)$, which is absolutely huge. Current

methods mainly use weight sharing strategy to boost the searching efficiency. In the typically fixed weight sharing pattern, each layer width $c_i \in [1 : n_i] = \{1, 2, ..., n_i\}$ is indicated by one and only one sub-network (see Fig.1(b)), and all layers within the sub-network adopt the same weight sharing pattern, then the size of search space $\mathcal{C}$ is reduced to $\prod_{i=1}^{L} n_i$.

After specifying each width with its corresponding sub-network, we can evaluate it by examining the performance of its sub-network in the supernet with shared weights. Then the whole searching consists of two steps, *i.e.*, supernet training and searching with supernet. Usually, the original training dataset is split into two datasets, *i.e.*, training dataset $\mathcal{D}_{tr}$ and validation dataset $\mathcal{D}_{val}$. The one-shot supernet $\mathcal{N}$ with weights $W$ is trained by randomly sampling a width $c$ and optimizing its corresponding sub-network with weights $\boldsymbol{w_c} \subset W$, *i.e.*,

$$W^* = \underset{\boldsymbol{w_c} \subset W}{\arg\min} \; \underset{\boldsymbol{c} \in U(\mathcal{C})}{\mathbb{E}} \left[ \mathcal{L}_{train}(\boldsymbol{w_c}; \mathcal{N}, \boldsymbol{c}, \mathcal{D}_{tr}) \right], \tag{1}$$

where $U(\mathcal{C})$ is a uniform distribution of network width in search space, and $\mathbb{E}[\cdot]$ is the expectation of random variables. After the supernet is trained, the performance (*e.g.*, classification accuracy) of various network width is indicated by evaluating its sub-network on the trained supernet $\mathcal{N}^*$. And the optimal width corresponds to that of the highest performance on the validation dataset, *e.g.*,

$$\boldsymbol{c}^* = \underset{\boldsymbol{c} \in \mathcal{C}}{\arg\max} \; \text{Accuracy}(\boldsymbol{w_c^*}, W^*; \mathcal{N}^*, \mathcal{D}_{val}), \; \text{s.t. FLOPs}(\boldsymbol{c}) \leq F_b, \tag{2}$$

where $F_b$ is a constraint on the FLOPs, here we consider FLOPs rather than latency as the hardware constraint since we are not targeting any specific hardware device like EfficientNet Tan & Le (2019) and other width search or channel pruning baselines Zhuang et al. (2018); You et al. (2020); Dong & Yang (2019); He et al. (2018b); Tang et al. (2020); Su et al. (2020). The searching of Eq.(2) can be fulfilled efficiently by various algorithms, such as random or evolutionary search (Hancock, 1992; Whitley et al., 1990; Liu et al., 2019a).

## 3 CafeNet

### 3.1 Locally free weight sharing

As illustrated before, a one-shot supernet $\mathcal{N}$ is usually trained as an evaluator to reflect the performance of different settings of network width. Assume we count channels in a layer from the left. To locate the sub-network in the supernet for width $c$ at a certain layer [2], conventional fixed weight sharing pattern assigns the leftmost $c$ channels as the sub-network as Fig.1(b). Formally, for width $c$, its corresponding sub-network is specified by a fixed *channel assignment* with an index set $\mathcal{I}(c)$ of channels, *i.e.*,

$$\text{fixed weight sharing pattern:} \quad \mathcal{I}(c) = [1 : c]. \tag{3}$$

However, this hard assignment imposes an inherent constraint on the search space. Channel assignment of smaller width will always be a subset of that of larger width. Actually, this fixed pattern forces the weight sharing of two different width to the greatest extent. To quantify this property, we define a simple weight sharing degree $d$ of two any width $c$ and $\tilde{c}$ by examining the overlap of their channel assignment, *i.e.*,

$$d(c, \tilde{c}) = \frac{|\mathcal{I}(c) \cap \mathcal{I}(\tilde{c})|}{\min(|\mathcal{I}(c)|, |\mathcal{I}(\tilde{c})|)} \in [0, 1]. \tag{4}$$

Intuitively, if the degree $d(c, \tilde{c})$ is larger, then sub-networks of width $c$ and $\tilde{c}$ will share more weights, and this implies that the supernet will enforce width $c$ and $\tilde{c}$ to have more similar performance. And for the fixed pattern, the weight sharing degree reaches its maximum as Theorem 1 [3].

**Theorem 1.** *The weight sharing degree of fixed pattern will always be 1 for any two width.*

However, ideally, each width should have full freedom in the supernet to select its channel assignment. In this situation, sub-networks of two different layer width can be fully shared, partially shared, or even have no overlap, thus the weight sharing degree freely varies from 0 to 1. And each width can be better evaluated by some sub-networks, instead of a manually assigned sub-network. Nevertheless,

---

[2] Here we omit the subscript $i$ for indicating any width $c_i$ at $i$-th layer.

[3] It holds naturally. Suppose $c \leq \tilde{c}$, then we have $\mathcal{I}(c) \cap \mathcal{I}(\tilde{c}) = \mathcal{I}(c)$ for fixed pattern.

this induces the search space to proliferate from the fixed $\mathcal{O}(n)$ to $\mathcal{O}(2^n)$, which is computationally unaffordable for practical search.

Therefore, we bridge these two extreme situations, and propose a locally free weight sharing strategy. Basically, to specify a sub-network of width $c$ from a supernet for evaluation, we need to assign $c$ channels from all $n$ channels per layer, with the channel assignment index set $\mathcal{I}(c)$. We split the index set $\mathcal{I}(c)$ into two parts, *i.e.*, $c_b$ base channels $\mathcal{I}_b(c)$ and $c_f$ free channels $\mathcal{I}_f(c)$, with $\mathcal{I}(c) = \mathcal{I}_b(c) \cup \mathcal{I}_f(c)$ and $c = c_b + c_f$. In detail, for free channels, we encourage to select $c_f$ channels from a given zone, which is the neighborhood of the $c$-th channel, *i.e.*, $\mathcal{B}(c; r) = [c - r : c + r]$, and $r$ is a preset allowed offset to control the search space with $c_f = r + 1$. Besides, base channels follow the fixed weight sharing pattern with $\mathcal{I}_b(c) = [0 : c_b]$; let $\mathcal{I}_b(c) \cap \mathcal{I}_f(c) = \emptyset$, then we can have $c_b = \max(c - r - 1, 0)$. In this way, in our formulation, we can specify multiple channel assignment for width $c$ to better evaluate its performance as Fig.1(b), *i.e.*,

$$\text{locally free weight sharing pattern:} \quad \mathcal{I}(c) = [1 : c_b] \cup \mathcal{I}_f(c), \text{ where } \mathcal{I}_f(c) \subset [c - r : c + r] \quad (5)$$

According to Eq.(4), the weight sharing degree of our method is between $c_b/c \le d(c, \tilde{c}) \le 1$. By properly setting offset $r$, we can achieve a tradeoff between the fixed pattern and the fully free pattern. Given a $r$, now our search space scales at $\mathcal{O}(\mathbb{C}_{2r+1}^{r+1} n)$, which is only a constant times (*e.g.*, $\mathcal{O}(3n)$ for $r = 1$, see Fig.1(b)) of fixed weight sharing search space. In this way, each width can be better represented, thus boosts the performance ranking ability of supernet.

**Training with min-min optimization.** With the above search space, we can train our CafeNet in a stochastic setting as Eq.(1), which simply samples a network width $c$ first, and then optimizes its sub-network. Nevertheless, in CafeNet a width is specified more freely by several sub-networks, and the performance of these sub-networks can be different. In this way, we propose to indicate the performance for each width by examining its sub-network with the best performance. In training CafeNet, instead of randomly optimizing a sub-network, we optimize the sub-network with the smallest training loss. To achieve this, we need to traverse all sub-networks for width $c$. However, this can be efficiently fulfilled since our job is to identify a sub-network with the minimum loss, which can be quickly calculated in a feed-forward manner. Afterward, we only need to implement a backward update for the target sub-network with minimum loss. A detailed explanation of this min-min optimization is elaborated in Appendix A.1.

**Searching with max-max selection.** After the CafeNet $\mathcal{N}$ is trained, we can evaluate each network width by examining its performance (*e.g.* classification accuracy) on the validation dataset $\mathcal{D}_{val}$. Similar to the training of CafeNet, for each width $c$, we use the sub-network with the highest performance to indicate the performance. Then we search for the optimal $c^*$ as Eq.(2). Note that since searching itself is much faster than the supernet training, the increased computational cost is subtle and acceptable in real practice. A detailed description of this max-max selection is presented in Appendix A.1.

## 3.2 Reducing search space with FLOPs-sensitive bins

Based on the locally free weight sharing pattern, each network width can be more flexibly indicated and then evaluated by the CafeNet. However, as previously illustrated, the size of the search space is considerably large and brings a tough challenge for the subsequent searching algorithms, such as random search and evolutionary search. In this way, current methods usually choose to reduce the search space by initializing the *minimum searching unit* as a group of channels (called *bin*) instead of a single channel. Concretely, all channels at each layer are partitioned into $K$ groups, with $(n_i/K)$ channels in each group for the $i$-th layer.

However, this uniform partition neglects the extra information of channels within layers, *i.e.*, kernel size and feature map resolution, and all of these factors are involved in the calculation of FLOPs. This inspires us that we should build bins based on the FLOPs. For simplicity, we encourage all bins in our search space to have the same FLOPs. Note that FLOPs is up to the layer width between two adjacent layers; for example, for convolution with resolution $H \times W$ and kernel size $K \times K$, its FLOPs can be calculated as

$$\text{FLOPs}(c_{in}, c_{out}) = c_{in} \times c_{out} \times H \times W \times K^2, \quad (6)$$

where $c_{in}$ and $c_{out}$ are the width for the input and output of this layer, respectively. In this way, channels at different layers contribute differently to the overall FLOPs of the supernet. And if a

channel at a layer contributes more to the FLOPs, then we can say it is more sensitive w.r.t. FLOPs. This *FLOPs-sensitivity* can be simply reflected by examining the real FLOPs of a single channel at a layer. Concretely, for each channel at the $i$-th layer, it serves as both the output channel of the $(i-1)$-th layer and input channel of the $(i+1)$-th layer. Thus the sensitivity $\varepsilon_i$ for the $i$-th layer can be represented as

$$\varepsilon_i = \text{FLOPs}(n_{i-1}, 1) + \text{FLOPs}(1, n_{i+1}), \tag{7}$$

where $n_{i-1}$ and $n_{i+1}$ are the width of the $(i-1)$-th and the $(i+1)$-th layer, respectively.

As a result, we simply encourage all bins to have the same FLOPs-sensitivity. To do so, we propose to initialize the bin size (*i.e.*, number of channels in a bin) more adaptively. The intuition is that the larger FLOPs sensitivity a layer has, the fewer channels should be allocated in a bin, and vice versa. As a result, the bin size $b_i$ for $i$-th layer should satisfy

$$b_i \propto 1/\varepsilon_i. \tag{8}$$

In practice, we usually specify a minimum bin size (*i.e.*, with maximum sensitivity $\varepsilon$) as $\beta$, then the bin size for each layer is

$$b_i = \beta \times \frac{\max_j \varepsilon_j}{\varepsilon_i}. \tag{9}$$

Actually, the minimum bin size $\beta$ in Eq.(9) controls the total number of bins, thus also determines the size of the search space. Specially, $b_i$ should be an integer and greater than or equal to 1. If $\beta$ is set to be smaller, the search space will be much larger and fine-grained since our search unit gets smaller and delicate accordingly. We will discuss a multi-stage search strategy by controlling $\beta$ actively in Section 3.3, and present empirical investigations in Appendix A.16.

### 3.3    FURTHER BOOSTING PERFORMANCE WITH MULTI-STAGE SEARCH

As previously illustrated, setting minimum bin size $\beta$ in Eq.(9) to be smaller will bring in a smaller searching unit, and thus a more fine-grained structure will be expected. However, this is accompanied by a larger search space and is challenging for both the ranking ability of supernet and searching performance. In this way, we can leverage a multi-stage searching strategy to avoid the search space being too large but still can search on a fine-grained level. In detail, we propose to search for the width under the multi-stage FLOPs budget, which is decayed linearly, *i.e.*,

$$\text{FLOPs}(t) \leftarrow \text{FLOPs}(0) - (\text{FLOPs}(0) - F_b) \times \frac{t}{T}, \quad \beta(t+1) \leftarrow \beta(t)/\alpha \tag{10}$$

where $\text{FLOPs}(t)$ represents the FLOPs budget in $t$-th stage and $\text{FLOPs}(0)$ is the FLOPs of the supernet. $F_b$ is the target FLOPs after all $T$ stages as Eq.(13). Besides, at each stage, we shrink the minimum bin size $\beta$ with evolving speed $\alpha > 0$. Thus the search space will evolve from the coarse-grained to the fine-grained, but with controllable size. More details are explored in Appendix A.14 and A.15. However, searching with the multi-stage strategy inevitably introduces more computation consumption. Therefore, we only implement this method in the small dataset (*i.e.* CIFAR-10).

## 4    EXPERIMENTAL RESULTS

In this section, we perform extensive experiments to validate the effectiveness of our CafeNet. For all architectures, we use the SGD optimizer with momentum 0.9. Parameters $\beta$ in Eq.(9) and $\alpha$ in Eq.(10) are set to 1 and 2, respectively. Offset $r$ for locally free channels is set to 1 for all experiments. Detailed experimental settings are elaborated in Appendix A.2.

We include various state-of-the-art width search methods for comparison, *e.g.* AutoSlim (Yu & Huang, 2019), MetaPruning (Liu et al., 2019a), TAS (Dong & Yang, 2019), GBN (You et al., 2019) and FPGM (He et al., 2019a). Besides, for comprehensive comparison, we perform searching using both evolutionary search and random search, named as **CafeNet-E** and **CafeNet-R**, respectively. In addition, we also consider two vanilla baselines, *i.e.*, Uniform and Random. Uniform: we shrink the width of each layer uniformly with a fixed factor to meet the FLOPs budget. Random: we randomly sample 20 networks under FLOPs constraint, and train them by 50 epochs, then we continue training the one with the highest performance and report its final result.

Table 1: Results on ImageNet of ResNet50 and MobileNetV2 compared with state-of-the-art methods. References of baseline methods are summarized in the Appendix A.4.

| ResNet50 | | | | | | MobileNetV2 | | | | | |
|---|---|---|---|---|---|---|---|---|---|---|---|
| | Methods | FLOPs | Param | Top-1 | Top-5 | | Methods | FLOPs | Param | Top-1 | Top-5 |
| 3G | AutoSlim | 3.0G | 23.1M | 76.0% | - | 200M | MetaPruning | 217M | - | 71.2% | - |
| | MetaPruning | 3.0G | - | 76.2% | - | | LEGR | 210M | - | 71.4% | - |
| | LEGR | 3.0G | - | 76.2% | - | | AutoSlim | 207M | 4.1M | 73.0% | - |
| | Uniform | 3.0G | 19.1M | 75.9% | 93.0% | | Uniform | 217M | 2.7M | 70.9% | 89.4% |
| | Random | 3.0G | - | 75.2% | 92.5% | | Random | 217M | - | 70.3% | 89.1% |
| | **CafeNet-R** | 3.0G | 22.6M | **77.1%** | 94.3% | | **CafeNet-R** | 217M | 3.0M | **73.3%** | 91.1% |
| | **CafeNet-E** | 3.0G | 23.8M | **77.4%** | 94.5% | | **CafeNet-E** | 217M | 3.3M | **73.4%** | 91.2% |
| 2G | GBN | 2.4G | 31.8M | 76.2% | 92.8% | 150M | LEGR | 180M | - | 70.8% | - |
| | LEGR | 2.4G | - | 75.7% | 92.7% | | TAS | 150M | - | **70.9%** | - |
| | FPGM | 2.4G | - | 75.6% | 92.6% | | AMC | 150M | - | **70.8%** | - |
| | TAS | 2.3G | - | 76.2% | 93.1% | | LEGR | 150M | - | 69.4% | - |
| | AutoSlim | 2.0G | 20.6M | 75.6% | - | | MuffNet | 149M | - | 63.7% | - |
| | Uniform | 2.0G | 13.3M | 75.1% | 92.7% | | Uniform | 150M | 2.0M | 69.3% | 88.9% |
| | Random | 2.0G | - | 74.6% | 92.2% | | Random | 150M | - | 68.8% | 88.7% |
| | **CafeNet-R** | 2.0G | 19.1M | **76.5%** | 93.1% | | **CafeNet-R** | 150M | 2.7M | **71.9%** | 90.0% |
| | **CafeNet-E** | 2.0G | 18.4M | **76.9%** | 93.3% | | **CafeNet-E** | 150M | 3.0M | **72.4%** | 90.4% |
| 1G | MetaPruning | 1.0G | - | 73.4% | - | 100M | MetaPruning | 105M | - | 65.0% | - |
| | AutoSlim | 1.0G | - | 74.0% | - | | Uniform | 105M | 1.5M | 65.1% | 89.6% |
| | Uniform | 1.0G | 6.6M | 73.1% | 91.8% | | Random | 105M | - | 63.9% | - |
| | Random | 1.0G | - | 72.2% | 91.4% | | **CafeNet-R** | 106M | 2.2M | **68.2%** | 88.2% |
| | **CafeNet-R** | 1.0G | 11.2M | **74.9%** | 92.3% | | **CafeNet-E** | 106M | 2.1M | **68.7%** | 88.5% |
| | **CafeNet-E** | 1.0G | 12M | **75.3%** | 92.6% | | MuffNet | 50M | - | 50.3% | - |
| 570M | AutoSlim | 570M | - | 72.2% | - | | MetaPruning | 43M | - | 58.3% | - |
| | Uniform | 570M | 4.0M | 71.6% | 90.6% | | Uniform | 50M | 0.9M | 59.7% | 82.0% |
| | Random | 570M | - | 69.4% | 90.3% | | Random | 50M | - | 57.4% | 81.2% |
| | **CafeNet-R** | 570M | 11.3M | **72.7%** | 90.9% | | **CafeNet-R** | 50M | 1.7M | **64.3%** | 85.2% |
| | **CafeNet-E** | 570M | 12.0M | **73.3%** | 91.2% | | **CafeNet-E** | 50M | 1.6M | **64.9%** | 85.4% |

Table 2: Searching results of EfficientNet-B0 with $1\times$ and $0.5\times$ FLOPs on ImageNet dataset.

| $1\times$ EfficientNet-B0 | | | | | $0.5\times$ EfficientNet-B0 | | | | |
|---|---|---|---|---|---|---|---|---|---|
| Methods | FLOPs | Param | Top-1 | Top-5 | Methods | FLOPs | Param | Top-1 | Top-5 |
| Uniform | 385M | 5.3M | 76.42% | 92.24% | Uniform | 192M | 2.8M | 74.16% | 91.48% |
| Random | 385M | 5.1M | 75.96% | 92.11% | Random | 192M | 3.0M | 73.36% | 91.03% |
| **CafeNet-R** | 385M | 6.2M | **76.59%** | 92.74% | **CafeNet-R** | 192M | 3.7M | **74.47%** | 91.65% |
| **CafeNet-E** | 385M | 6.9M | **76.83%** | 92.96% | **CafeNet-E** | 192M | 3.9M | **74.62%** | 91.87% |

## 4.1 RESULTS ON IMAGENET DATASET

We perform experiments on ResNet50 (He et al., 2016), MobileNetV2 (Sandler et al., 2018) and EfficientNet-B0 (Tan & Le, 2019) to examine the performance of CafeNet on heavy and light models, as summarized in Table 1 and 2. In detail, ResNet50 and MobileNetV2 are searched from the original size, and for EfficientNet-B0, the $0.5\times$ FLOPs is also searched from the original model. But the $1.0\times$ FLOPs is from a supernet with $1.5\times$ FLOPs from uniform width scaling. The original EfficientNet-B0, ResNet50 and MobileNetV2 has 5.3M, 25.5M and 3.5M parameters and 386M, 4.1G and 300M FLOPs with 76.4%, 77.8% and 73.7% Top-1 accuracy, respectively. Note that we also adopt KD in the final training of MobileNetV2 for fair comparison with baseline methods, *i.e.*, AutoSlim (Yu & Huang, 2019), TAS (Dong & Yang, 2019). More detailed experiments and visualization results of our searched network are shown in Appendix A.8 and A.11.

As shown in Table 1, our CafeNet achieves the highest performance on ResNet50 and MobileNetV2 w.r.t. different FLOPs, which proves the superiority of our method compared to other width search algorithms. Besides, CafeNet obtains remarkable performance with tiny FLOPs budgets. For example, our 570M ResNet50 achieves 73.3% Top-1 accuracy, surpassing AutoSlim (Yu & Huang, 2019) by 1.1%. In addition, as shown in Table 2, for the NAS-based network EfficientNet-B0, the performance can be further improved using CafeNet. It indicates that our CafeNet can optimize its width in a more fine-grained manner.

## 4.2 RESULTS ON CIFAR-10 DATASET

We also investigate CafeNet with MobileNetV2 (Sandler et al., 2018) and VGGNet (Simonyan & Zisserman, 2014) on the moderate CIFAR-10 dataset. Our original VGGNet (MobileNetV2) has 20M (2.2M) parameters and 399M (297M) FLOPs with accuracy of 93.99% (94.81%). As shown in Table 3, our CafeNet enjoys significant superiority to other algorithms. In detail, our 44M MobileNetV2 achieves 95.31% accuracy and outperforms MuffNet (Chen et al., 2019) by 2.2%. For VGGNet,

Table 3: Results of MobileNetV2 and VGGNet on CIFAR-10 dataset. References of baseline methods are summarized in Appendix A.4.

| | | MobileNetV2 | | | | | VGGNet | | | |
|---|---|---|---|---|---|---|---|---|---|---|
| Groups | Methods | FLOPs | Parameters | Accuracy | Groups | Methods | FLOPs | Parameters | Accuracy |
| 100M+ | DCP | 218M | 1.7M | 94.75% | 200M | GAL | 190M | - | 93.80% |
| | **CafeNet-R** | 188M | 1.4M | **95.44%** | | DCP | 199M | 10.4M | 94.16% |
| | **CafeNet-E** | 188M | 1.5M | **95.56%** | | Sliming | 199M | 10.4M | 93.80% |
| | MuffNet | 175M | - | 94.71% | | **CafeNet-R** | 189M | 8.3M | **94.27%** |
| | **CafeNet-R** | 144M | 1.2M | **95.28%** | | **CafeNet-E** | 189M | 8.0M | **94.36%** |
| | **CafeNet-E** | 144M | 1.1M | **95.44%** | 100M+ | PS | 156M | - | 93.63% |
| 100M- | AutoSlim | 88M | 1.5M | 93.20% | | **CafeNet-R** | 154M | 3.4M | **94.09%** |
| | AutoSlim | 59M | 0.7M | 93.00% | | **CafeNet-E** | 154M | 3.1M | **94.23%** |
| | MuffNet | 45M | - | 93.12% | | AOFP | 124M | - | 93.84% |
| | **CafeNet-R** | 44M | 0.4M | **95.16%** | | **CafeNet-R** | 115M | 2.4M | **93.87%** |
| | **CafeNet-E** | 44M | 0.4M | **95.31%** | | **CafeNet-E** | 115M | 2.1M | **94.01%** |
| | AutoSlim | 28M | 0.3M | 92.00% | 76M | CGNets | 92M | - | 92.88% |
| | **CafeNet-R** | 28M | 0.2M | **93.87%** | | **CafeNet-R** | 76M | 2.2M | **93.36%** |
| | **CafeNet-E** | 28M | 0.2M | **94.11%** | | **CafeNet-E** | 76M | 1.4M | **93.67%** |

Table 4: Results of MobileNetV2 and ResNet18 on CelebA dataset.

| | MobileNetV2 | | | | ResNet18 | | | |
|---|---|---|---|---|---|---|---|---|
| FLOPs | 21M | 51M | 106M | 162M | FLOPs | 130M | 316M | 619M | 1G |
| Uniform | 91.63% | 91.73% | 91.92% | 91.97% | Uniform | 91.66% | 91.79% | 91.93% | 92.03% |
| Random | 91.42% | 91.52% | 91.63% | 91.76% | Random | 91.51% | 91.62% | 91.67% | 91.84% |
| **CafeNet-R** | **91.71%** | **92.03%** | **92.09%** | **92.12%** | **CafeNet-R** | **91.83%** | **92.07%** | **92.13%** | **92.17%** |
| **CafeNet-E** | **91.85%** | **92.13%** | **92.16%** | **92.19%** | **CafeNet-E** | **91.92%** | **92.16%** | **92.18%** | **92.25%** |

Table 5: Results of ResNet50 and MobileNetV2 on MS COCO dataset (Lin et al., 2017b;a).

| | ResNet50 | | | | MobileNetV2 | | |
|---|---|---|---|---|---|---|---|
| Framework | Original $1.0\times$ | CafeNet $0.5\times$ | Uni $0.5\times$ | Framework | Original $1.0\times$ | CafeNet $0.5\times$ | Uni $0.5\times$ |
| RetinaNet | 36.3% | 34.9% | 34.2% | RetinaNet | 31.2% | 29.2% | 27.5% |
| Faster R-CNN | 37.2% | 36.0% | 35.5% | Faster R-CNN | 31.7% | 29.1% | 28.5% |

our CafeNet achieves 94.36% accuracy with 189M FLOPs, which surpasses DCP (Zhuang et al., 2018) by 0.2%. Similarly, our CafeNet shows great advantages w.r.t. tiny models, *e.g.* with 2.11% accuracy improvement than AutoSlim (Yu & Huang, 2019) on MobileNetV2 with 28M FLOPs. More experiments on CIFAR-10 dataset are investigated in Appendix A.9.

## 4.3 EXPERIMENTS ON FACE ATTRIBUTE TASK WITH CELEBA DATASET

To futher investigate the effectiveness of CafeNet, we perform experiments of multi-label classification task on CelebA (Liu et al., 2015) dataset with MobileNetV2 (Sandler et al., 2018) and ResNet18 (He et al., 2016). CelebA dataset has 162K training and 20K testing images with 40 attribute labels from 2 categories. We report the average accuracy of all 40 labels. The original MobileNetV2 (ResNet18) has 213M (1.3G) FLOPs and 2.2M (11.2M) parameters with 92.07% (92.13%) average accuracy.

As shown in Table 4, our CafeNet can even achieve higher accuracy with FLOPs reducing a bit. For example, our 162M-FLOPs MobileNetV2 has 92.19% average accuracy and outperforms the original model by 0.12%. Besides, our algorithm still enjoys great superiority in tiny FLOPs; for example, our 130M-ResNet18 achieves 91.92% average accuracy, which is 0.26% higher than the uniform baseline. Besides, to comprehensively compare the classification performances on all 40 labels, we present the comparison results for each label in Appendix A.9.

## 4.4 TRANSFERABILITY OF THE SEARCHED WIDTH TO OBJECT DETECTION TASK

For object detection tasks, its backbone is usually initialized by a pretrained model on ImgeNet dataset. In this way, we use MobileNetV2 and ResNet50 with $0.5\times$ FLOPs as the backbone feature extractors to examine the transferability of our searched network width. Besides, we leverage both the two-stage Faster R-CNN with Feature Pyramid Networks (FPN) (Lin et al., 2017a; Ren et al., 2015) and the one-stage RetinaNet (Lin et al., 2017b) frameworks for verification. We train models using the *trainval* split of MS COCO (Everingham et al., 2010) as training data and report the results in mean Average Precision (Lin et al., 2017a; Ren et al., 2015) on *minival* split. As shown in Table 5, the backbones obtained by our method (CafeNet $0.5\times$) can achieve higher performance compared to the uniform baseline (Uni $0.5\times$).

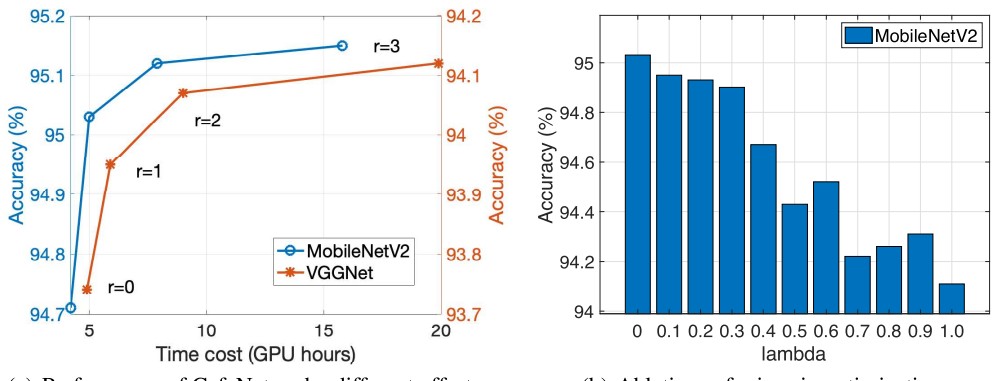

(a) Performance of CafeNet under different offsets $r$      (b) Ablations of min-min optimization

Figure 2: (a) Performance with different offset $r$ of free channels for the 0.5×-FLOPs MobileNetV2 and VGGNet on CIFAR-10 dataset. (b) Ablations of min-min optimization with 0.5×-FLOPs MobileNetV2 on CIFAR-10 dataset.

## 4.5 Ablation Studies

**Effect of offset $r$ for free channels.** As described before, free channels freely locate in the local zone indicated by an offset $r$. Since we search on bins, $r$ controls the size of search space; larger $r$ implies more accurate evaluation of width but induces larger search space, corresponding to more computational cost. To investigate its effect, we compare the performance and time cost between different $r$ on 0.5×-FLOPs MobileNetV2 and VGGNet on CIFAR-10 dataset. As shown in Fig.2(a), accuracy performance benefits from the increase of $r$ while the larger $r$ also aggravates the burden of training. For a trade-off between performance and time cost, we set $r = 1$ in all experiments. Note that in the training of CafeNet, larger $r$ will have more sub-networks; however, identifying the one with the minimum loss only involves feed-forward calculation and does not increase training cost dramatically. More results refer to Appendix A.5.

**Effect of min-min optimization on supernet**. During training CafeNet, we only optimize the sub-network with minimum training loss for each sampled width. To investigate the effect of this optimization strategy, suppose we have all $\tau$ iterations, then we implement min-min optimization only on the last $(1 - \lambda) \cdot \tau$ iterations with $\lambda \in [0, 1]$. For the first $\lambda \cdot \tau$ iterations, we simply optimize one sub-network randomly for each sampled width. As shown in Fig.2(b), our 0.5×-FLOPs MobileNetV2 on CIFAR-10 improves 0.92% accuracy from $\lambda = 1$ to $\lambda = 0$, which means that min-min optimization does help to better evaluate each width and boost the searching performance accordingly. We further record the performance of 1K sampled width; details refer to Appendix A.13.

**Effect of performance improvement with more training and searching cost**. Since CafeNet needs extra training and searching cost than the fixed weight sharing pattern, one natural question comes to *whether a fixed pattern can benefit from more training and searching cost and even surpass CafeNet as a result*. With this aim, we implement a fixed pattern (*i.e.*, $r = 0$ for CafeNet) with more times (1×∼3×) of training epochs and generations using evolutionary search to search for a 0.5×-FLOPs MobileNetV2 and VGGNet on CIFAR-10 dataset.

Table 6: Performance with more training and searching cost on CIFAR-10 dataset. Note CafeNet has 95.44% and 94.36% accuracy for 0.5×-FLOPs MobileNetV2 and VGGNet, respectively.

| MobileNetV2 | | | | VGGNet | | | |
|---|---|---|---|---|---|---|---|
| Searching \ Training | 1× | 2× | 3× | Searching \ Training | 1× | 2× | 3× |
| 1× | 94.53% | 94.57% | 94.55% | 1× | 93.62% | 93.65% | 93.66% |
| 2× | 94.61% | 94.67% | 94.69% | 2× | 93.68% | 93.73% | 93.71% |
| 3× | 94.72% | 94.75% | 94.74% | 3× | 93.69% | 93.66% | 93.74% |

As shown in Table 6, we can see that searching with more generations by evolutionary search can only improve the searching results slightly, while more times of training cost for supernet has almost no effect on the searching performance. By comparing with Table 3 and Fig.2(a), our CafeNet does efficiently improve the performance of searching results with the same or a little extra training and searching cost.

## 5   CONCLUSION

In this paper, we introduce a new weight sharing pattern for network width search. In detail, our locally free pattern enables each width is jointly indicated by its base channels and free channels, where free channels are used to better and more flexibly to evaluate the performance. Besides, we leverage FLOPs-sensitive bins to reduce the search space and allow FLOPs to distribute more evenly for different layers. CafeNet can be trained stochastically using min-min optimization and searches the optimal width by max-max selection. Extensive experiments have been conducted on benchmark ImageNet, CIFAR-10, CelebA, and COCO datasets to show the superiority of our CafeNet to other state-of-the-art methods.

### ACKNOWLEDGMENTS

This work is funded by the National Key Research and Development Program of China (No. 2018AAA0100701) and the NSFC 61876095. Chang Xu was supported in part by the Australian Research Council under Projects DE180101438 and DP210101859.

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
