# OpenReview forum: "Locally Free Weight Sharing for Network Width Search"
_ICLR.cc/2021/Conference — ICLR 2021 Spotlight_

### Official Review · AnonReviewer1 · 2020-10-23
**An interesting weight sharing mechanism for network width search**

**Rating:** 8
**Confidence:** 4

**Review:**

In this paper, the authors introduce a new weight sharing pattern to search for the width in a network layer. Besides, FLOPs-sensitive bins is proposed to measure the real FLOPs of a single channel at a layer and further reduce the search space. The paper proposes a locally free weight sharing mechanism where the channels in a layer are split into base channels and free channels. Compared with conventional fixed weight sharing pattern where the leftmost channels are assigned as the sub-network, the proposed locally free pattern increases more flexibility while the search space also scales at O(n). The proposed FLOPs-sensitive bins forces the layers with larger FLOPs sensitivity to have fewer channels, thus reducing the search space at a fixed FLOPs. Experimental results on several datasets show that the proposed CafeNet outperforms many other width search algorithms. The searched network experimentally achieves high performance with tiny FLOPs budgets.

What I like about this paper in that:
1.	The motivation and intuition are reasonable, which is to design a more flexible weight sharing pattern for network width search.
2.	Experiments are sufficient, thorough and carefully designed. Experimental results can support the objective of proposed methods. The searched network achieves remarkable performance with tiny FLOPs budgets.
3.	The paper is well written and organized. The work is easy to follow and be reproduced.
4.	The proposed methods have high generality and might be used on any convolutional network.

Some minor concerns or suggestion about this paper:
1.	The searching and training algorithms (max-max selection and min-min optimization) should be described in more detail.
2.	The free channels are the neighborhood of the c-th channel in this paper, but I think more channels on the right should be included in the zone.

---

> ### Author Response · Authors · 2020-11-20
> **Response to AnonReviewer1**
>
> Thanks for the reviewer's effort on reviewing our paper. The responses to the reviewer’s questions are as follows:
>
> Q1: Explanation of max-max selection and min-min optimization. \
> A1:  With the proposed locally free weight sharing pattern, width is specified more freely by several sub-networks, and the performance of these sub-networks can be different. Therefore, to directly compare the performance of different network width, we need to specify an indicator for each width. For example, use the Top-1 accuracy of all sub-networks (CafeNet) or other methods that involve more sub-networks(e.g., the average accuracy of all sub-networks). Although there are countless ways to specify the indicator, they will undoubtedly introduce more computation for each width. Sub-networks with poor performance may not sufficiently represent the performance of width. Therefore, to indicate the performance w.r.t network width, we propose to leverage the sub-network with maximum performance, which amounts to the minimum loss. (min-min optimization)
>
> Besides, during the search, for evaluating each network width $c$, we follow a similar max-max selection strategy by leveraging the sub-network with the highest performance to indicate its performance.
>
> Q2: The assignment of free channels. \
> A2: Thanks for your valuable idea. In this paper, we propose to leverage a more freely assigned weights pattern for network weights. Suppose we use both the locally free weight pattern and assign channels more freely(i.e., channels on the right side), which may cause an unfair comparison with the current mainstream baseline methods (algorithms with fixed weight pattern). However, with more freely assigned channels, network width search performance may be further increased, which can be researched as innovative work in the future.

---

### Official Review · AnonReviewer2 · 2020-10-28
**Official Blind Review #2**

**Rating:** 6
**Confidence:** 5

**Review:**

Most existing methods follow a manually ﬁxed weight sharing pattern, leading to the difficulty that estimates the performance of networks with different widths. To address this issue, this paper proposes a locally free weight sharing strategy (CafeNet) to share weights more freely. Moreover, this paper further proposes FLOPs-sensitive bins to reduces the size of the search space. Specifically, this paper divides channels into several groups/bins that have the same FLOPs-sensitivity and searches for promising architectures based on the divided groups. Extensive experiments on several benchmark datasets demonstrate superiority over the considered methods. However, some important details regarding the proposed method are missing. My detailed comments are as follows.

Positive points:
1. Compared with the manually ﬁxed weight sharing pattern, this paper proposes a locally free weight sharing strategy (CafeNet), which allows more freedom in the channel assignment of a sub-network.

2. To reduce the size of the search space, this paper proposes to divide channels into several groups/bins (also called minimum searching unit) that have the same FLOPs-sensitivity.

3. The experimental results on image classification and object detection tasks show that the proposed method outperforms the existing methods by a large margin.

Negative points:
1. When training the super network, why the authors optimize the sub-network with the smallest training loss? More explanations are required.

2. Why the sensitivity of a layer should be calculated as Eqn. (7)? It would be better to provide more details about that.

3. Given a FLOPs constraint in Eq. (2), how to select a suitable width for each layer? Please discuss more and make it clearer.

4. Is it possible to find a sub-network with zero width ($c=0$) for a layer? If so, how to deal with this case when evaluating the sub-network?

5. The experimental results are inconsistent with the descriptions. In Figure 2(b), the performance of the proposed method goes worse with the decreasing of the $\lambda$. However, the authors state that “MobileNetV2 on CIFAR-10 improves 0.92% accuracy from $\lambda$ =0 to $\lambda$=1”.

6. The experimental comparisons in Table 1 are unfair. Compared with other methods (e.g. AutoSlim), the proposed method trains the models on ImageNet for more epochs (100 v.s. 300). More experiments under the same settings are required.

Minor issues:
1. In appendix A.13, “… and the bin evolving speed α in Section 3.4” should be “… and the bin evolving speed α in Section 3.3”.

---

> ### Author Response · Authors · 2020-11-20
> **Response to AnonReviewer2**
>
> Thanks for your positive support and instructive comments. We have revised the presentation issue and typos of this paper in our next version.
>
> Q1: More explanations with the smallest training loss. \
> A1: Since network width is specified more freely by several sub-networks in CafeNet, and the performance of these sub-networks can be different. Therefore, to directly compare the performance of different network width, we need to specify an indicator for each network width. For example, use the Top-1 accuracy of all sub-networks (CafeNet) or other methods that involve more sub-networks(e.g., the average accuracy of all sub-networks). Although there are countless ways to specify the indicator, these ways will undoubtedly introduce more computation for each width, and also sub-networks with poor performance may not well represent the performance of width. Therefore, to indicate the performance w.r.t network width, we propose to leverage the sub-network with maximum performance, which amounts to the smallest training loss.
>
> Q2: More details of Eq. (7). \
> A2: Since the FLOPs of a layer is linear to the number of filters(channels), to evaluate the influence of filters, we propose to construct the FLOPs-sensitivity of a layer by examining the real FLOPs variation of reducing a single filter. In detail, a filter influences the output channel for the current layer and the input channel as the next layer, which corresponds to two terms of Eq. (7).
>
> Q3: Selection of layer width with Eq. (2). \
> A3: In our method, for the given FLOPs constraint, the search is implemented with random or evolutionary search, named CafeNet-R and CafeNet-E in all tables. In detail, for evolutionary search, we implement it with the multi-objective NSGA-II algorithm [1]. As illustrated in Appendix A.2, we set the population and iteration size of evolutionary search to 40 and 50, and we randomly select 40 network width within the FLOPs budget as the initial population. In each generation,  with each width satisfying the FLOPs budget, we specify the sub-network with the strategy of max-max selection to indicate its performance. Then, we assign the validation accuracy (the validation dataset is split from the training dataset) of the specified sub-network for the width to indicate its performance. For those sampled network width with larger FLOPs, we just drop them. Afterward, the network width with the highest score is selected as the optimal width to train from scratch for evaluation.
> While for the random search, we uniformly sample the same number of network width as the evolutionary search. Then, for each network width satisfying the FLOPs budget, we specify the sub-network with the strategy of max-max selection and examine its validation accuracy. Afterward, we select the width with the highest accuracy to train from scratch for evaluation. \
>  [1] Deb, Kalyanmoy, et al. "A fast and elitist multiobjective genetic algorithm: NSGA-II." IEEE transactions on evolutionary computation 6.2 (2002): 182-197.
>
> Q4. Zero width issue for a layer. \
> A4: The current setting of CafeNet cannot reach 0 width for two reasons. First, as described in Eq. (5), we limit the width of each layer to no less than 1. Second, even if 0 width can be selected as a candidate, the performance of the corresponding sub-network will be greatly restricted due to the existence of the disconnected layer, and thus cannot be selected as the optimal width through evolution or random search.
>
> Q5. Presentation issue of Fig. 2(b). \
> A5: Thanks for pointing out this issue. In fact, $\lambda$ in the text corresponds to $1 - \lambda$ in Fig 2(b). Therefore, some descriptions of min-min optimization should be revised as follows:
> During training CafeNet, we only optimize the sub-network with minimum training loss for each sampled width. To investigate the effect of this optimization strategy, suppose we have all $\tau$ iterations, then we implement min-min optimization only on the last $(1-\lambda)\cdot\tau$ iterations with $\lambda\in[0,1]$. For the first $\lambda\cdot\tau$ iterations, we simply optimize one sub-network randomly for each sampled width. As shown in Fig. 2(b), our 0.5$\times$-FLOPs MobileNetV2 on CIFAR-10 improves 0.92\% accuracy from $\lambda = 1$ to $\lambda = 0$.
>
> Q6. More experiments of searched network width with aligned hyperparameter settings. \
> A6: Thanks for your advice. As illustrated in Appendix A.10, we retrain the searched network width with the same training recipes of AutoSlim, as shown in Table. 14. Some examples of the comparisons are shown below: \
> 1G FLOPs of ResNet50: \
> AutoSlim: 74.0\%, CafeNet: 74.5\% \
> 207M FLOPs of MobileNetV2: \
> AutoSlim: 73.0\%, CafeNet:  73.2\% \
> More details about the retraining results can refer to as Table. 14 of Appendix A.10. With the same training recipes of the baseline method (AutoSlim), the results in Appendix A.10 show the effectiveness of our proposed CafeNet.

---

### Official Review · AnonReviewer4 · 2020-10-28
**An interesting idea with convincing results**

**Rating:** 8
**Confidence:** 4

**Review:**

This paper explores the weight sharing schema in one-shot width search and proposes a locally free weight sharing strategy (CafeNet). By splitting each width candidate into base channels and free channels, CafeNet makes a compromise between fixed weight pattern and full freedom pattern. Such strategy can reduce the search complexity and improve the performance ranking, w.r.t. different width, in the supernet. Experiments on various tasks, including classification, detection and attribute recognition, are well provided to support the effectiveness of the proposed method. The final results are quite promising.

Strengths:
1) The paper is well written and easy to follow. The motivation is clearly explained by an example and the problem formulation.
2) The idea of locally free weight sharing is interesting. Such a solution for the previous fixed weight sharing seems sound.
3) Experiments with additional analyses are well provided.

I have the following concerns and suggestions:
1) Missing some relevant papers. OFA[1] and TF-NAS[2] introduce width search by dynamically choosing the channels. The authors should cite and explain the differences.
2) Is there any correlation between the degree in Eq. (4) and the searched accuracy under a fixed FLOPs?
3) The bin size of Eq. (9) makes me confusing. As shown in experiments, β can be less than 1 and the second term in the right side of Eq. (9) is also less than 1. Thus, the bin size bi (i.e., number of channels in a bin) is less than 1 channel. Please explain it in detail.
4) Why lambda=0 achieves the best accuracy in Fig. 2(b)? It is in conflict with the statement “As shown in Fig. 2(b), our 0.5-FLOPs MobileNetV2 on CIFAR-10 improves 0.92% accuracy from lambda=0 to lambda=1”.
5) I suggest the authors to split 1G group in Table 1, as done in Table 11.
6) In Algorithm1, both the supernet and the total number of epoch are defined as N.

Although some details make me a little confusing, the experimental analyses and the intuitive solutions of locally free weight sharing in one-shot width search are quite informative and helpful to the NAS community. I suggests the authors to release their code and would like to see the authors’ responses.


[1] Han Cai, Chuang Gan, Tianzhe Wang, Zhekai Zhang, Song Han. Once-for-All: Train One Network and Specialize it for Efficient Deployment. ICLR, 2020.

[2] Yibo Hu, Xiang Wu, Ran He. TF-NAS: Rethinking Three Search Freedoms of Latency-Constrained Differentiable Neural Architecture Search. ECCV, 2020.

---

> ### Author Response · Authors · 2020-11-20
> **Response to AnonReviewer4**
>
> Thanks for your positive opinions and suggestions. We have revised the typos and polished our presentation according to the response below.
>
> Q1: Discussion of related papers. \
> A1: Thanks for reminding of the related works. OFA proposes to train a supernet that supports diverse architectural settings by decoupling training and search; thus, it can be quickly used to get a specialized sub-network without additional training. TF-NAS proposes a novel method to boost the search with three levels of differentiable NAS (i.e.,  operation-level, depth-level, and width-level).  With this method,  TF-NAS achieves good performance in terms of both classification accuracy and precise latency constraint.  We have cited these papers (i.e., OFA and TF-NAS) in the first paragraph of introduction.
>
>
> Q2: Correlation of weight sharing degree with performance under the same FLOPs. \
> A2: Indeed, higher degrees of freedom will lead to better search results. As shown in Figure 2(a), with the same FLOPs budget, accuracy performance benefits from the increase of $r$. In detail, when $r$ is set to 1, a large gap is introduced in comparison to $r = 0$ (fixed weight pattern); this is because a better representation for each width is induced with the freedom in selecting channels.  However, when $r$ goes larger, the increase of performance of searched width gradually slows down, which means using a small offset $r$ can already help distinguish the performance of different width, thus helping to select the optimal width.
>
> Q3: Typo of Eq. (9). \
> A3: The Eq. (9) should be revised to $ b_i = \beta \times \frac{\max_j \varepsilon_j}{\varepsilon_i}$.  Thanks for pointing out this typo. Since we specify the minimum bin size as $\beta$, and bin size is inverse to the sensitivity $\varepsilon$ as defined in Eq. (8). Therefore, the minimum bin size should correspond to the maximum sensitivity.  With this revised Eq. (9),  the second term on the right side of Eq. (9)  will always be $\geq$ 1. As for the definition of $\beta$, in practical implementation, the number of channels in each bin should be an integer and greater than or equal to 1. Therefore, in the code level, the bin size $b_i$ should be implemented as $b_i = round(\max(b_i, 1))$. As a result, $\beta \leq 1$ means that the bin size of several layers with $\varepsilon$ close to the maximum sensitivity $\max_j \varepsilon_j$ are defined to 1, which induces a larger search space.
>
> Q4: Presentation issue of Fig. 2(b). \
> A4: Sorry for the inconsistent meanings of lambda. In fact, $\lambda$ in the text corresponds to $1 - \lambda$ in Fig 2(b). Therefore, the descriptions of min-min optimization should be revised as follows:   \
> During training CafeNet, we only optimize the sub-network with minimum training loss for each sampled width. To investigate the effect of this optimization strategy, suppose we have all $\tau$ iterations, then we implement min-min optimization only on the last $(1-\lambda)\cdot\tau$ iterations with $\lambda\in[0,1]$. For the first $\lambda\cdot\tau$ iterations, we simply optimize one sub-network randomly for each sampled width. As shown in Fig. 2(b), our 0.5$\times$-FLOPs MobileNetV2 on CIFAR-10 improves 0.92\% accuracy from $\lambda = 1$ to $\lambda = 0$, which means that min-min optimization does help to better evaluate each width and boost the searching performance accordingly. We further record the performance of 1K sampled width; details refer to Appendix A.13.
>
>
> Q5: Code release and other presentation suggestions: \
> A5: Thanks for your suggestions. We will release our code after this paper is published, and we have modified the presentation of Table 1 and use $\mathcal {E}$ (Epochs) to indicate the epochs in algorithm1.

---

### Official Review · AnonReviewer3 · 2020-10-29
**Interesting Attempt on Network Width Search**

**Rating:** 7
**Confidence:** 4

**Review:**

The authors introduce in this submission a locally-free weight sharing strategy for selecting effective network width. The rationale and intuition behind are well-grounded. Experiments on various datasets and pruning setups prove the validity.

Strength:
+ The approach is well motivated and makes sense. The problem studied here is also important and could be of interest to a large audience.
+ Experiments are sufficient. The results are promising and well support the claim.
+ FLOPs-sensitivity bin considers factors including feature size and kernel size and seems to be independent of the total channel number, which, without douts, brings values.

Weakness:
- The proposed approach seems to be a  compromise between completely free weight and fixed weight, right? As a result, it would be good if the authors could elaborate the relation between the two.
- By utilizing the methods, my understanding is that the search space scales from O(N) all the way up to O(C_{2r+1}^{N}), no? This is a considerable amount of time required as compared to the single network width. Please provide some discussion along this line.
- The influence of the super network should be detailed. Intuitively, higher degrees of freedom will lead to better results. The authors should provide more analysis along this line.
- The writing can be enhanced. Please go over the manuscript and make sure all the grammar errors have been taken care of.

---

> ### Author Response · Authors · 2020-11-20
> **Response to AnonReviewer3**
>
> Thanks for the comments. We have polished our writing, and the following answers have been revised accordingly in our next version.
>
> Q1: The relation between completely free weight and fixed weight. \
> A1: For fixed weight pattern, it simply assigns the left $c$ channels as the sub-network for the width $c$. However, for the completely free weight pattern, to assign layer width of $c$ from $l$ channels in a layer, there will be $\mathbb{C}^c_l$ kinds of configurations of weights, which is computationally unaffordable for practical search. Therefore, we bridge these two extreme situations by proposing a locally free weight sharing (CafeNet) strategy. In CafeNet, we split the channels of a layer into two parts, i.e., base channels and free channels, with base channels following the strategy of fixed weight sharing pattern while free channels from the neighborhood of the $c$-th channel with a preset allowed offset of $r$. The search space of our method scales at $\mathcal{O}(\mathbb{C}_{2r+1}^{r+1}n)$, which is only a constant time of (e.g., $\mathcal{O}(3n)$ for $r = 1$) of fixed weight sharing search space.
>
> Q2: The size of search space. \
> A2: With the proposed locally free weight pattern, the size of the search space is indeed scaled from  $\mathcal{O}(n)$ to $\mathcal{O}(\mathbb{C}_{2r+1}^{N})$. Instead of randomly sampling and optimizing the sub-networks, we focus on the sub-network with the best performance for a particular width. In other words, each width only corresponds to one sub-network, which largely enhances the efficiency. Details about this strategy of min-min optimization can be found in the Appendix with Eqs. (11-13). A similar strategy is applied during the evaluation of sub-networks as well.
>
> Q3: Analysis of freedom within CafeNet. \
> A3: It is indeed that higher degrees of freedom will lead to better results. As shown in Figure 2(a), accuracy performance benefits from the increase of $r$ (more freedom). In detail, when $r$ is set to 1, a large gap is introduced in comparison to $r = 0$(fixed weight pattern); this is because a better representation for each width is induced with the free channels. However, when $r$ grows larger, the increase of searched width performance gradually slows down, which means using a small offset $r$ can already help distinguish the performance of different width. Thus a larger offsets $r$ can only lead to a little performance improvement with the selected width.  Nevertheless, the larger $r$ also aggravates the burden of training as Table 6. For a trade-off between performance and time cost, we set $r = 1$ in all experiments.
>
> Q4: Writing issue.  \
> A4: We carefully proofread this paper and fix the typos.

---

### Decision · Program_Chairs · 2021-01-07
**Final Decision**

**Decision:**

Accept (Spotlight)

**Comment:**

The paper proposed locally free weight sharing strategy (CafeNet) for searching optimal network width. The proposal is a nice tradeoff between manually fixed weight sharing pattern (too small search space) and completely free weight sharing pattern (too large search space). The *originality* and *significance* are clearly above the bar. The paper is related to the general interests of deep learning research and its *applicability* deserves a spotlight presentation.

It seems the *clarity* can still be improved, so please carefully revise the paper following the reviews. BTW, I am very curious, why "locally free weight sharing strategy" goes to a short name CafeNet? I went over the paper but I didn't find the answer. Perhaps the name of the proposal should also be explained...